# Optimal Deployment of Wireless Charging Infrastructure for Electric Tram with Dual Operation Policy

Young Kwan Ko [1], Yonghui Oh [2], Dae Young Ryu [1] and Young Dae Ko [1,*]

1 Department of Hotel and Tourism Management, College of Hospitality and Tourism, Sejong University, 209 Neungdong-ro, Gwangjin-gu, Seoul 05006, Korea; kyk0423@sju.ac.kr (Y.K.K.); rdy6255@sju.ac.kr (D.Y.R.)
2 Department of Industrial and Management Engineering, Daejin University, 1007, Hoguk-ro, Pocheon-si 11159, Gyeonggi-do, Korea; oryong@daejin.ac.kr
* Correspondence: youngdae.ko@sejong.ac.kr; Tel.: +82-10-4725-3480

**Abstract:** The wireless charging electric tram system is presently receiving attention as an eco-friendly means of transportation. The conventional electric tram system has a similar advantage in regards to environmental pollution, but it has several problems that are caused by the overhead power supply line. The battery-type electric tram system should be considered carefully, because the battery itself is an environmentally harmful material. Therefore, the wireless charging electric tram system is regarded as an alternative means of transportation. The adequate battery capacity and the location of the wireless charging infrastructure are investigated in this study, which consider the dual operation policy, and the objective is to minimize the total investment cost. The variation of the battery capacity and the location of the wireless charging infrastructure are examined that compare Case 1, which involves the electric trams operating only in normal operations, and Case 2, which includes the electric trams operating in normal and express operations.

**Keywords:** wireless charging technology; electric tram; transportation policy; battery capacity; wireless power transfer





## 1. Introduction

The usage of wireless charging technology has been extended to areas, such as cellular phones, vacuum cleaners, and other electric devices that need to be charged or provided with electricity in real-time. This trend is also applied to the electric vehicle industry, which is due to its conveniences. As a result, the on-line electric vehicle (OLEV), which is a wireless charging electric vehicle technology, was developed by the Korea Advanced Institute of Science and Technology (KAIST), Korea in 2009, and it was successfully installed for use in the Gumi, Korea, which is shown in Figure 1 [1]. In addition, OLEV bus have been commercialized since 2019 in Daejeon Metropolitan City, Korea, and are operated with actual customers paying for them.

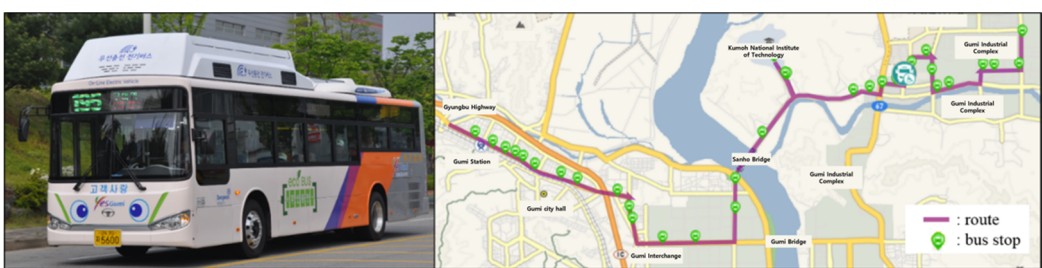

**Figure 1.** OLEV bus and operation route installed at Gumi city.

The wireless charging technology can also be applied to electric trams in a similar way to the OLEV bus, forming a wireless charging electric tram scheme. The wireless

charging electric tram system is regarded as an innovative alternative means of mass transportation in urban and suburban areas. In Taiwan, the wireless charging electric tram was piloted over an 8.2 km section in 2014. The conventional electric tram system has several critical problems, which are due to the overhead power supply line in terms of aesthetical, maintenance, and safety perspectives. In some congested sections such as intersections, overhead power lines are inevitably installed in a messy manner. In order to remove this overhead power supply line, one of the alternatives is a battery-type electric tram system. However, the investment cost is very high, because it requires a long charging down time, and the price of a battery is very expensive. In addition, the battery is environmentally harmful material. However, with the wireless charging electric tram, electricity is charged via a wireless charging infrastructure that is buried under the railway even though it is moving. This means that when the wireless charging electric tram is operated on the railway where the wireless charging infrastructure is installed, electricity can be provided wirelessly from the wireless charging infrastructure, which is illustrated on the left part of Figure 2. However, when the wireless charging electric tram is operated on the railway where the wireless charging infrastructure is not installed, it should then consume the electricity that is stored in its equipped battery, which is shown on the right part of Figure 2. Therefore, it is possible to operate certain routes with a relatively small battery capacity as well as save the charging down time. In addition, wireless charging electric trams can be operated with a smaller battery capacity, thereby reducing the use of batteries that have a harmful effect on the environment.

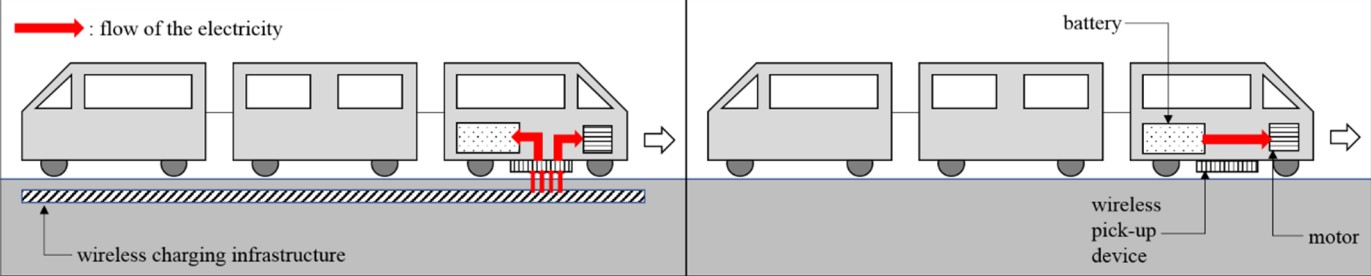

**Figure 2.** Electricity flow of the wireless charging electric tram system.

One of the purposes of mass transportation is to transfer the passengers to their destinations as fast as possible. In order to cope with this situation efficiently, several policies tend to be applied for mass transportation systems, such as the bus rapid transit and transfer systems among the other means of mass transportation. In this study, the dual operation policy for the electric tram system is investigated by considering the features of the wireless charging technology. The dual operation policy describes the different types of operations among the trams, which are operated on the same route. Suppose that there is an express operation and a normal operation for the trams along the same routes. The express electric tram stops at several selected stations, whereas the normal electric tram stops at all the stations. Figure 3 depicts the operation routes of Metro 9 in Seoul Metropolitan City, Korea. There is a total of 30 stations, and 12 stations are set as express stations. The normal electric tram stops at every station for Metro 9, whereas the express electric tram only stops at the express stations. Therefore, passengers who want to travel from Gimpo Airport to Yeouido can reach their destination more quickly by taking an express electric tram as opposed to taking a normal electric tram.

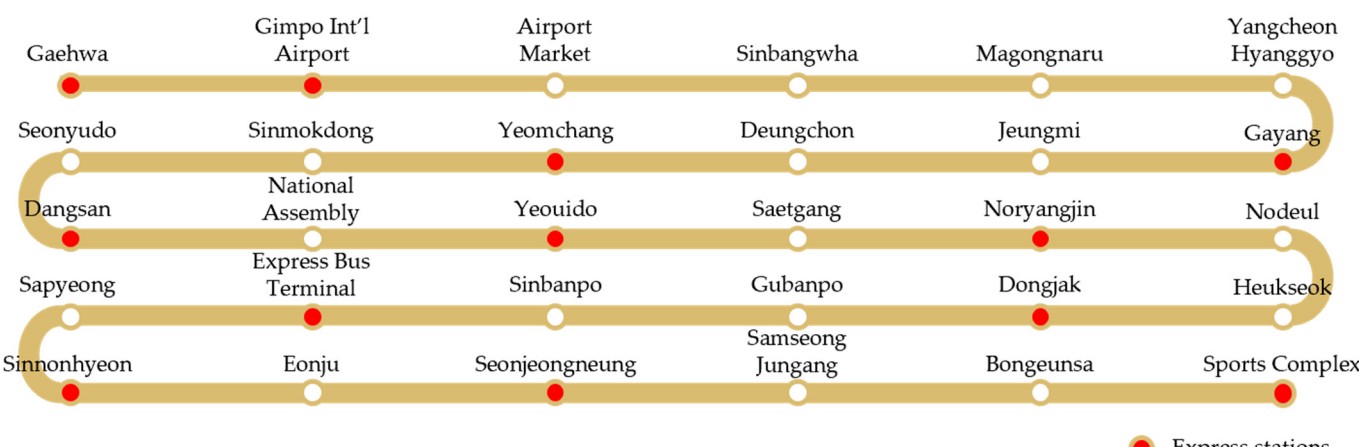

**Figure 3.** Operation routes for Metro 9 of Seoul Metropolitan City, Korea.

The express tram generally stops at the stations where many people hop on and off. However, if there are too many express stations compared to normal stations, the dissatisfaction of the customers that use normal stations can increase. Therefore, the decision about the location of the express stations is one of the important issues in this field. However, it beyond the scope of this study, because the focus here is the optimal deployment of the wireless charging electric tram system that considers both the express and normal operation policies. This means that the variations of the decision-making elements about the required battery capacity and the allocation of the wireless charging infrastructure will be observed and examined that compare Case 1 and Case 2. Case 1 deals the situation where electric trams operate only with normal operations, and Case 2 deals the case of the electric trams that operate with normal and express operations.

As such, this study tries to derive the optimal design of a wireless charging electric tram system with minimal investment cost for successful commercialization. For that, it is investigated to find both the optimal value of battery capacity and optimal allocation of wireless charging infrastructure for wireless charging electric tram through mathematical model-based optimization. In addition, this study does not simply consider the one-time operation of the wireless charging electric tram, but also reflect the actual operation policies such as the normal line, express line and so on.

This paper is organized as follows. In Section 2, the previous related studies are presented as the literature review. A description of the nature of the problem, which includes the dual operation policy for the wireless charging electric tram system and the mathematical model for the wireless charging electric trams with a dual operation policy, in order to minimize the total investment cost are explained and developed in Section 3. The computational experiments and sensitivity tests with the different operation ratios between the express and the normal operations are introduced in Section 4. Finally, the findings and insights from this study are presented as concluding remarks in Section 5.

## 2. Literature Review

Interest in the electric vehicle system has increased as an alternative means of transportation since 2010 in order to reduce environmental pollution. At the same time, interest in the wireless charging electric vehicle system has been raised in order to overcome the two main weaknesses of conventional electric vehicle systems, which include long charging down time and expensive battery. Therefore, there is a great deal of research about the wireless charging electric vehicle systems, but most of the research discussed the development with technology aspects, which are not applications with managerial aspects. Ko and Jang [2] introduced the optimal system design of the On-Line Electric Vehicle, which is one type of the wireless charging electric vehicles that was developed by KAIST in 2009. They considered the minimization of the total investment cost in order to deploy the wire-

less charging electric vehicle systems on certain routes as well as deciding the minimum required battery capacity and the location of the wireless charging infrastructure. They extended their research with a focus on the allocation of the wireless charging infrastructure, which included continuous or separate, and applied several solution generation techniques, such as a meta-heuristics such as genetic algorithm, particle swarm optimization, and commercial solution software CPLEX [3,4]. Hwang et al. [5] recently derived an optimal system design of the wireless charging electric vehicle system for complex multiple routes. It is hard to consider the general model about the allocation of the wireless charging infrastructure with multiple routes. This is because the overall routes can be divided or merged at lots of points and complex situation at these points, such as intersections. In addition, Liu and Song [6] addressed the design of the optimal battery sizes and the location of the wireless charging infrastructure for a dynamic wireless power transfer electric bus system. They proposed the methodology of robust optimization in order to deal with the uncertainty in regards to the energy consumption and the travel time. Vaz et al. [7] discussed the optimal acceleration of electric vehicles where the previous research did not consider the acceleration duration. They presented a multi objective optimization problem in order to minimize both the acceleration duration and the energy consumption. The suggested problem was solved using the multi objective genetic algorithm. In addition, Jang et al. [8] explained an initial investment cost analysis for the wireless charging electric vehicles when it is applied to a public transportation system. They considered three different types of the wireless charging systems, such as stationary wireless charging, quasi-dynamic wireless charging, and dynamic wireless charging. The authors also performed a cost-sensitivity analysis. Chen et al. [9] discussed the allocation of two types of charging facilities, which include charging stations for stationary charges and charging lanes for dynamic charges along a long traffic corridor. They analyzed the charging-facility-choice equilibrium of electric vehicles among the locations and two types of charging facilities. Nahum and Hadas [10] developed an optimal allocation model for wireless charging stations of electric bus. They suggested a non-linear optimization model with multiple objectives; (1) minimize the costs, (2) maximize the environmental benefit, (3) minimize the number of charging stations. Genetic algorithm is used to solve the presented problem and numerical tests to verify the fit of the model were also performed.

The express operation in this study is referred to as the skip-stop operation in the literature. The skip-stop operation is one of the most effective ways to increase the operation speed of the transportation type, such as a train or bus. The main advantage of the skip-stop operation is to reduce the passenger travel time, which includes the waiting times, in-vehicle travel times, and transfer times, without any additional facility investment [11]. There have been several studies that deal with the skip-stop operation in the rail transportation industry. Lee et al. [12] developed a mathematical model using a genetic algorithm, and they decided to stop and skip stations for the skip-stop rail operation. As a result of the simulation with Seoul Metro line 4, the total travel time was about 17–20 percent shorter than the original all-stop operation. Suh et al. [13] also studied the skip-stop operation in the Seoul Metro system. They found that when the skip-stop operation was applied to line 5, the total travel time decreased, but the waiting time increased compared to the all-stop system. Abdelhafiez et al. [14] suggested the skip-stop arrangement in Urban Rail Transit System (URT) in order to minimize the average travel time of the passengers. A heuristic algorithm was developed to solve the large size problems, and 25 sample problems were generated and tested in order to verify the performance of the proposed heuristic algorithm. Niu et al. [15] provided an adjusted train timetable with a given predetermined train skip-stop pattern in order to minimize the total waiting time of the passengers. They developed a unified quadratic integer model with linear constraints, and the numerical experiments of a real-world rail train timetable were solved using a GAMS (General Algebraic Modeling System). Zhang et al. [16] proposed a new skip-stop operation methodology, which is called the flexible skip-stop scheme (FSSS), that can provide more combinations of stop modes in order to satisfy the passenger demand. They

aimed to minimize the total travel time as well as maximize the number of passengers during the off-peak hours. Wang et al. [17] considered a real-time train scheduling problem under a skip-stop operation with the objective of minimizing the total travel time of the passengers and the energy consumption of the operation of trains. They developed new iterative convex programming (ICT) and compared it with nonlinear programming (NLP), which is a mixed-integer nonlinear programming (MINLP), and a mixed-integer linear programming (MILP). Yang et al. [18] dealt with the train stopping planning and the train scheduling problems at the same time in order to minimize the total dwelling time and the total delay between the real and the expected departure time. GAMS with a CPLEX solver was used to solve the proposed model, and two sets of numerical examples were implemented in order to verify the performance. Jiang et al. [19] addressed the train scheduling problem with the objective of increasing the number of scheduled trains that were provided in the currently scheduled timetable. In order to satisfy the increasing passenger demand, it was considered to increase the dwelling time or to skip a few stops. A heuristic algorithm was developed and tested using the Chinese high-speed Jing Hu corridor, which runs between Beijing and Shanghai.

However, there is insufficient research that concerns wireless charging electric trams with an operation policy for efficient transportation, because it is a newly introduced means of transportation. Therefore, the dual operation policy with the express electric tram, which can stop at several selected stations, and the normal electric tram, which would stop at all stations is considered in this study. In order to consider sustainable transportation, the wireless charging electric tram system is considered.

## 3. Model Development

### 3.1. Problem Description

The wireless charging electric tram system described in Figure 2 can be considered as one of the possible alternatives where conventional trams are operating. However, it is necessary to make an optimal decision on the installation of an appropriate battery capacity and wireless charging infrastructure for the successful installation and operation of this eco-friendly system. Battery capacity is necessary to ensure that the tram does not completely discharge considering that it uses the electric energy of the battery while operating the entire route and receives electric energy from the wireless charging infrastructure. In addition, wireless charging infrastructure requires one inverter for a series of inductive cables, so the overall investment can vary depending on the installation design. Therefore, the optimal decision-making algorithm for battery capacity and wireless charging infrastructure installation should be developed.

Suppose that the wireless charging electric tram system is applied at certain regions to reduce the emission of air pollution materials, and the dual operation policy is also adopted in order to provide the efficient transportation service for the customers. The decision-making elements of this study under this situation are the required battery capacities of both the express and the normal electric trams as well as whether the wireless charging infrastructure are allocated or not at each segment on the overall railway system.

It is assumed for an easier application that the overall railway system is divided by *I* number of separated regions, and each region is called as segment. The decision that concerns where to allocate the wireless charging infrastructure can then be considered at each segment level. It is further assumed that the geographic information, such as the number and location of the stations and distance between the consecutive stations are known, whereas the physical information, such as the mass of the wireless charging electric tram, the coefficient of friction, and the aerodynamic drag are also known. Moreover, it is assumed that the operation profile of the wireless charging electric tram, such as acceleration, deceleration, and maximum velocity is predetermined.

The wireless charging infrastructure contains both an inverter and an inductive cable, and each linked inductive cable requires one inverter. Suppose that there is one linked inductive cable that is 100 m, which only requires one inverter. However, suppose that

there are two separated inductive cables that are 40 m and 60 m, two inverters for the two separated inductive cables are now required.

Based on these assumptions, the dual operation policy, which includes both express and normal electric trams for the wireless charging electric tram system, are examined in order to observe the different behavior of the decision-making elements, which include the required battery capacities for both the express and normal electric trams and the allocation of the inverter and the inductive cable at each segment.

### 3.2. Notations

A mathematical model-based optimization technique is applied in this study in order to derive the optimal values for the decision-making elements. Therefore, several notations are defined as follows in order to develop the mathematical model, which are provided below.

| **Indices and Index Sets** | | |
|---|---|---|
| $i$ | : | Set of segments, which the overall railway is divided by $I$ number of segments ($i = 1, 2, 3, \ldots, I$) |
| $k$ | : | Types of operations of the electric tram when $k = 1$, which means the normal operation of the electric tram, whereas $k = 2$ represents the express operation of it ($k = 1, 2$) |
| **Parameters** | | |
| $c_{tram}$ | : | Unit purchasing cost of the wireless charging electric tram without a battery [\$] |
| $c_{battery}$ | : | Unit purchasing cost of a battery per kWh [\$/kWh] |
| $c_{inverter}$ | : | Unit purchasing cost of the inverter [\$] |
| $c_{cable}$ | : | Unit purchasing cost of the inductive cable per meter [\$/meter] |
| $N_k$ | : | The number of type $k$ electric trams |
| $w$ | : | The maximum number of segments which can be covered by unit inverter |
| $b$ | | Battery capacity of unit battery back [kWh] |
| $l$ | : | Length of the unit segment [meter] |
| $L$ | : | Length of the overall railway [meter] |
| $t_i$ | : | Elapsed time passing $i$th segment [second] |
| $T$ | : | Overall elapsed time [second] |
| $\alpha_{max}$ | : | Maximum utilization ratio of a battery considering rapid charging, $0 \leq \alpha \leq 1$ |
| $\alpha_{min}$ | : | Minimum utilization ratio of a battery considering safety, $0 \leq \beta \leq 1$ |
| $d_{i,k}$ | : | An amount of consumed electricity at the $i$th segment for the type $k$ electric tram [kWh] |
| $s_{i,k}$ | : | An amount of supplied electricity by the wireless charging at the $i$th segment for the type $k$ electric tram [kWh] |
| $r_{i,k}$ | : | An amount of supplied electricity by the regenerative braking at the $i$th segment for the type $k$ electric tram [kWh] |
| **Decision variables** | | |
| $x_i$ | : | Binary decision variable, which has a value of 1 when the inductive cable is allocated at the $i$th segment, otherwise, it has value of 0 |
| $y_i$ | : | Binary decision variable, which has value of 1 when the inverter is allocated at the $i$th segment, otherwise, it has value of 0 |
| $q_{battery,k}$ | : | Minimum required battery capacity for the wireless charging electric tram, which is operated by type $k$ [kWh] |
| $I_{max,k}$ | : | Maximum utilization level of a battery for the type $k$ electric tram [kWh] |
| $I_{min,k}$ | : | Minimum utilization level of a battery for the type $k$ electric tram [kWh] |
| $I_{i,k}$ | : | Battery charging level after passing the $i$th segment for the type $k$ electric tram [kWh] |

### 3.3. Mathematical Model

The purpose of the mathematical model is to generate an optimal system design of the wireless charging electric tram system for the dual operation policy, which minimizes the total investment cost. Therefore, the objective function of the proposed mathematical model should be a minimization of the total related cost by deriving the adequate values of

the decision variables, such as the battery capacities for both the normal and the express electric trams and the allocation of both the inverter and the inductive cable at each segment. Therefore, the objective function can be modelled, which is shown in Equation (1).

Minimize,

$$\sum_{k=1}^{K} \left( c_{tram} + c_{battery} \cdot q_{battery,k} \right) \cdot N_k \\ + c_{inverter} \cdot \sum_{i=1}^{I} y_i + c_{cable} \cdot \sum_{i=1}^{I} l \cdot x_i \tag{1}$$

The objective function contains the purchasing cost of the type *k* electric tram with a battery, the purchasing cost of all the installed inverters, and the purchasing cost of all the installed inductive cables. Note that the purchasing cost of the type *k* electric tram with a battery and the purchasing cost of all the installed inverters and inductive cables have a trade-off relationship.

Subject to,

$$I_{max,k} = \alpha_{max} \cdot q_{battery,k}, \ \forall k \tag{2}$$

$$I_{min,k} = \alpha_{min} \cdot q_{battery,k}, \ \forall k \tag{3}$$

$$I_{min,k} \leq I_{i,k} \leq I_{max,k}, \ \forall k, \forall i \tag{4}$$

Suppose that the battery capacity of the type *k* electric tram is decided as $q_{battery,k}$, then the maximum and the minimum utilization level of a battery of the type *k* electric tram can be calculated using Equations (2) and (3). The battery charging level after passing the *i*th segment always within that range can be determined using Equation (4).

$$I_{0,k} = I_{max}, \ \forall k \tag{5}$$

$$I_{i-1,k} - d_{i,k} + r_{i,k} + s_{i,k} \cdot x_i \geq I_{i,k}, \forall k, \ \forall i \tag{6}$$

The initial battery charging level of the type *k* electric tram is set as its maximum utilization level in Equation (5). The battery charging level after passing the *i*th segment is equal or less than the summation of the battery charging level after passing the $(i-1)$th segment, the consumed electricity at *i*th segment, and the supplied electricity by the wireless charging and the regenerative braking, in which is illustrated with Equation (6).

$$x_0 = 0 \tag{7}$$

$$y_i \geq x_i - x_{i-1}, \ \forall i \tag{8}$$

Equation (7) is used for the initial condition of the allocation of the inductive cable, and Equation (8) can be used to determine whether the inverter is installed in the *i*th segment or not according to whether the inductive cable of the *i*th segment and the $(i-1)$th segment is installed.

$$\sum_{i=j}^{j+w} x_i \leq w, \ j = 1, \ldots, i - w \tag{9}$$

Equation (9) is developed to present the maximum number of segments covered by unit inverter. Since one inverter cannot cover an inductive cable longer than a certain length, this constraint has been proposed.

$$q_{battery,k} \geq 0, \ \forall k \tag{10}$$

$$x_i, y_i \in \{0, 1\}, \ \forall i \tag{11}$$

Finally, Equations (10) and (11) determine the non-negativity constraints. The battery capacities of the type *k* electric tram should be zero or a positive number, whereas the decision variables that indicate whether an inverter and an inductive cable are installed in the *i*th segment or not are the 0-1 binary variables.

*3.4. Solution Procedure*

The proposed mathematical model is developed using the linear form. Therefore, it can be solved in a conventional mathematical way. In addition, by applying the commercial optimal solution generation software, such as CPLEX, the optimal solution can be obtained within a reasonable time frame. Therefore, the values of the battery capacities for the type *k* electric trams and the values of the number of the installed inverters and inductive cables will be generated by CPLEX. Then it will be compared considering both Case (1), while the electric trams operate only with normal operation, and Case (2), while the electric trams operate with normal and express operations.

## 4. Computational Experiments

*4.1. Preparation for Experiments*

A numerical experiment is conducted in order to check the consistency of the model that was devised in this study. This research considers the wireless charging electric tram, so the actual data for a subway is applied. Therefore, it is assumed that the wireless charging electric tram system is adopted for the Metro 9 subway train in the Seoul Metropolitan City. There are 30 stations in the overall route, and the length of it is 31.7 km. It is currently operated with a dual operation policy, such as the express operation such as skip-stop and the normal operations, which includes stopping at all stations. The location and distance from the first station of each station is very important in order to interpret the results, which are described in Table 1. Note that the underlined station is the station where the wireless charging electric tram for express operation stop.

**Table 1.** Station information for Metro 9 in Seoul Metropolitan City.

| Station Number and Name | Distance from the 1st Station |
|---|---|
| 1. Gaehwa * | 0 m |
| 2. Gimpo Intl Airport * | 3600 m |
| 3. Airport Market | 4400 m |
| 4. Sinbanghwa | 5200 m |
| 5. Magongnaru | 6100 m |
| 6. Yangcheon Hyanggyo | 7500 m |
| 7. Gayang * | 8800 m |
| 8. Jeungmi | 9500 m |
| 9. Deungchon | 10,500 m |
| 10. Yeomchang * | 11,400 m |
| 11. Sinmokdong | 12,300 m |
| 12. Seonyudo | 13,500 m |
| 13. Dangsan * | 14,500 m |
| 14. National Assembly | 16,000 m |
| 15. Yeouido * | 16,900 m |
| 16. Saetgang | 17,700 m |
| 17. Noryangjin * | 18,900 m |
| 18. Nodeul | 20,000 m |
| 19. Heukseok | 21,100 m |
| 20. Dongjak * | 22,500 m |
| 21. Gubanpo | 23,500 m |
| 22. Sinbanpo | 24,200 m |
| 23. Express Bus Terminal * | 25,000 m |
| 24. Sapyeong | 26,100 m |
| 25. Sinnonhyeon * | 27,000 m |
| 26. Eonju | 27,800 m |
| 27. Seonjeongneung * | 28,700 m |
| 28. Samseong Jungang | 29,500 m |
| 29. Bongeunsa | 30,300 m |
| 30. Sports Complex * | 31,700 m |

* stations for express operation.

In order to calculate the battery consumption according to the operation and the battery charging by both the wireless charging and the regenerative braking, the power modeling proposed by Ko and Jang [20] is applied. Several parameters for power modeling are defined as Table 2.

**Table 2.** System parameters for power modeling.

| | Parameter | Value |
|---|---|---|
| $\mu_{rr}$ | Coefficient of rolling resistance | 0.02 |
| $A$ | Front area of the vehicle [m$^2$] | 4 |
| $m$ | Mass of unit OLEV [kg] | 120,000 |
| $C_d$ | Coefficient of the aerodynamic drag | 1.25 |
| $\psi$ | Climbing angle | 0 |
| $\varepsilon$ | Efficiency | 0.8 |
| $\rho$ | Air density [kg/m$^3$] | 1 |
| $g$ | Acceleration of gravity [m/s$^2$] | 9.8 |
| $P_{ac}$ | Required extra electrical power for heating, lighting, etc. [kWh] | 100 |

The operation profile for both the express tram and the normal tram are also given parameters. The accelerations of both the express tram and the normal tram when they start and stop at certain station are set as 1 m/s$^2$ and $-1$ m/a$^2$, respectively. In addition, the maximum velocity of both the express tram and the normal tram is set as 20 m/s. Moreover, the amount of the charging electricity by the wireless charging is set as 600 kWh/s, but it is set relatively higher, because it is not a small vehicle but a tram. The remaining system parameters for the problem solving are illustrated in Table 3. Note that the cost of tram vehicle is assumed as zero because the number of trams is fixed in this study.

**Table 3.** System parameters for problem solving.

| Parameter | Value | Parameter | Value | Parameter | Value |
|---|---|---|---|---|---|
| $c_{vehilce}$ | \$0/unit | $c_{battery}$ | \$1000/kWh | $c_{inverter}$ | \$50,000/unit |
| $c_{cable}$ | \$200/meter | $\alpha_{max}$ | 0.8 | $\alpha_{min}$ | 0.2 |
| $l$ | 50 m | $w$ | 40 | $b$ | 5 kWh |

This paper examines and compares when the wireless charging electric tram is operated only using the normal operation, which is Case 1, and when the wireless charging electric tram is operated using the express and the normal operation simultaneously, which is Case 2, based on those system parameters. The variations of the decision variables, such as the required minimum battery capacities and the allocations of the wireless charging infrastructure can then be observed, and several insights derived from those comparisons are expected.

*4.2. General Results*

First of all, the results for the two cases are displayed in Tables 4 and 5 when Case 1 is a situation when the wireless charging electric tram is operated only by using the normal operation, whereas Case 2 is a situation when the wireless charging electric tram is operated by using the express and the normal operation simultaneously. Case 1 deals with 20 times of normal operation, and Case 2 deals with 10 times of normal operation and 10 times of express operation. It is further assumed that the normal tram in Case 2 should be stopped at Gayang (7th station), Yeouido (15th station), and Sinnonhyeon (25th station) in order to avoid the express tram for 3 min at each station.

**Table 4.** Results of the computational experiments (Case 1).

| Parameter | Case 1-1 | Case 1-2 |
|---|---|---|
| | Normal Tram | Normal Tram |
| Number of trams | 20 unit | 20 unit |
| Battery capacity | 28.30 kWh | 30.00 kWh |
| Battery cost(all) | $566,000 | $600,000 |
| Number of inverters | 13 | 13 |
| Inverter cost | $650,000 | $650,000 |
| Location of the inductive cable (Number of allocated segments) | 1–39 (39)<br>41–79 (39)<br>86–125 (40)<br>142–180 (39)<br>187–216 (30)<br>225–259 (35)<br>267–304 (38)<br>318–357 (40)<br>373–411 (39)<br>419–458 (40)<br>467–505 (39)<br>520–559 (40)<br>573–612 (40) | 1–32 (32)<br>35–74 (40)<br>76–110 (35)<br>118–157 (40)<br>174–213 (40)<br>225–255 (31)<br>262–300 (39)<br>318–357 (40)<br>366–405 (40)<br>419–458 (40)<br>467–505 (39)<br>520–559 (40)<br>571–610 (40) |
| Number of total allocated segments | 498 | 496 |
| Inductive cable cost | $4,980,000 | $4,960,000 |
| Total investment cost | $6,196,000 | $6,210,000 |

**Table 5.** Results of the computational experiments (Case 2).

| Parameter | Case 2-1 | | Case 2-2 | |
|---|---|---|---|---|
| | Express Tram | Normal Tram | Express Tram | Normal Tram |
| Number of trams | 10 unit | 10 unit | 10 unit | 10 unit |
| Battery capacity | 252.38 kWh | 65.97 kWh | 255.00 kWh | 70.00 kWh |
| Battery cost(all) | $2,523,800<br>$3,183,500 | $659,700 | $2,550,000<br>$3,250,000 | $700,000 |
| Number of inverters | 11 | | 11 | |
| Inverter cost | $550,000 | | $550,000 | |
| Location of the inductive cable (Number of allocated segments) | 1–14 (14)<br>69–108 (40)<br>119–156 (38)<br>174–192 (19)<br>209–248 (40)<br>287–294 (8)<br>319–356 (38)<br>375–402 (28)<br>448–486 (39)<br>498–524 (27)<br>539–578 (40) | | 1–4 (4)<br>69–106 (38)<br>120–158 (39)<br>167–192 (26)<br>210–232 (23)<br>269–293 (25)<br>319–356 (38)<br>375–402 (28)<br>447–472 (26)<br>484–523 (40)<br>538–577 (40) | |
| Number of total allocated segments | 331 | | 327 | |
| Inductive cable cost | $3,310,000 | | $3,270,000 | |
| Total investment cost | $7,043,500 | | $7,070,000 | |

In this study, the battery capacities installed on both electric trams are defined as real numbers. However, in reality, commercial battery packs can be installed. In that case, the battery capacity is determined by how many battery packs are installed. Therefore, '-1' is appended to the case for calculating the minimum required battery capacity (in real number), and '-2' is appended to the case in which it is expressed as the number of battery packs (multiple of a certain number). In this study, the capacity of unit battery pack (=$b$) is assumed as 5 kWh.

It is observed that the total investment cost in the case of operating 10 express trams and 10 normal trams (Case 2-1) is more expensive than operating only 20 normal trams (Case 1-1). This means that almost 13.68% additional cost is required in order to apply the dual operation policy in this example. Of course, the amount of the additional cost can be derived differently according to the system circumstances, such as the various cost structure, the operation ratio between the express and normal trams, and the number of operations for both the express and normal trams. In Case 1-1, the ratio of the wireless charging infrastructure cost per total investment cost is $5,630,000/$6,196,000 = 0.91, whereas in Case 2-1, it is only $3,860,000/$7,043,500 = 0.55. This is because both the express tram and the normal tram tend to apply relatively higher battery capacity in order to cope with the different operation profiles when the express tram and the normal tram are operated at the same time (Case 2-1). However, the normal tram tries to apply a relatively smaller capacity battery when only the normal tram is operated on this route (Case 1) because it is easier to allocate the wireless charging infrastructure only for the normal tram, which is due to same operation profile.

Comparing 1-1 and 1-2 (or 2-1 and 2-2), the total investment cost of 1-2 (2-2) is larger than that of 1-1 (2-1). This is because the battery capacity can only be set in multiples of a certain number due to the assumption that a battery pack is used. Although the length of the required inductive cable may be slightly reduced due to the battery capacity installed more than the actual required amount, it can be observed that the cost is a little more in terms of the overall cost. However, since the difference in the value is not large, it is expected that the difference will be negligible no matter which method is used in actual field.

It takes a minimum of several tens of seconds to a maximum of several hours to derive the optimal solution of the MIP developed in this study through CPLEX. When the value of the system parameter changes, the calculation time also varies greatly, because the battery capacity and the wireless charging infrastructure have a sensitive trade-off relationship with each other.

The variation of the battery charging level during their operations in both cases is illustrated in Figure 4. The *x*-axis of graph means the route that the electric tram uses, whereas the *y*-axis denotes the battery level of the electric tram. The grey area indicates the segments that installed the wireless charging infrastructure, whereas the black curve represents the variations of the battery charging level during the operations in Case 1-1 and Case 2-1. Therefore, when the black curve passes the gray area, the value increases and decreases repeatedly, which is illustrated in Figure 4. This means the battery is charged in that area, whereas the battery decreases in the white area. The reason the black curve decreases at the chargeable area is because it consumes energy in order to move. The electric tram charges the battery at the white area, and this is due to the regenerative braking. Note that there are maximum and minimum utilization areas of the battery, which are due to its characteristics.

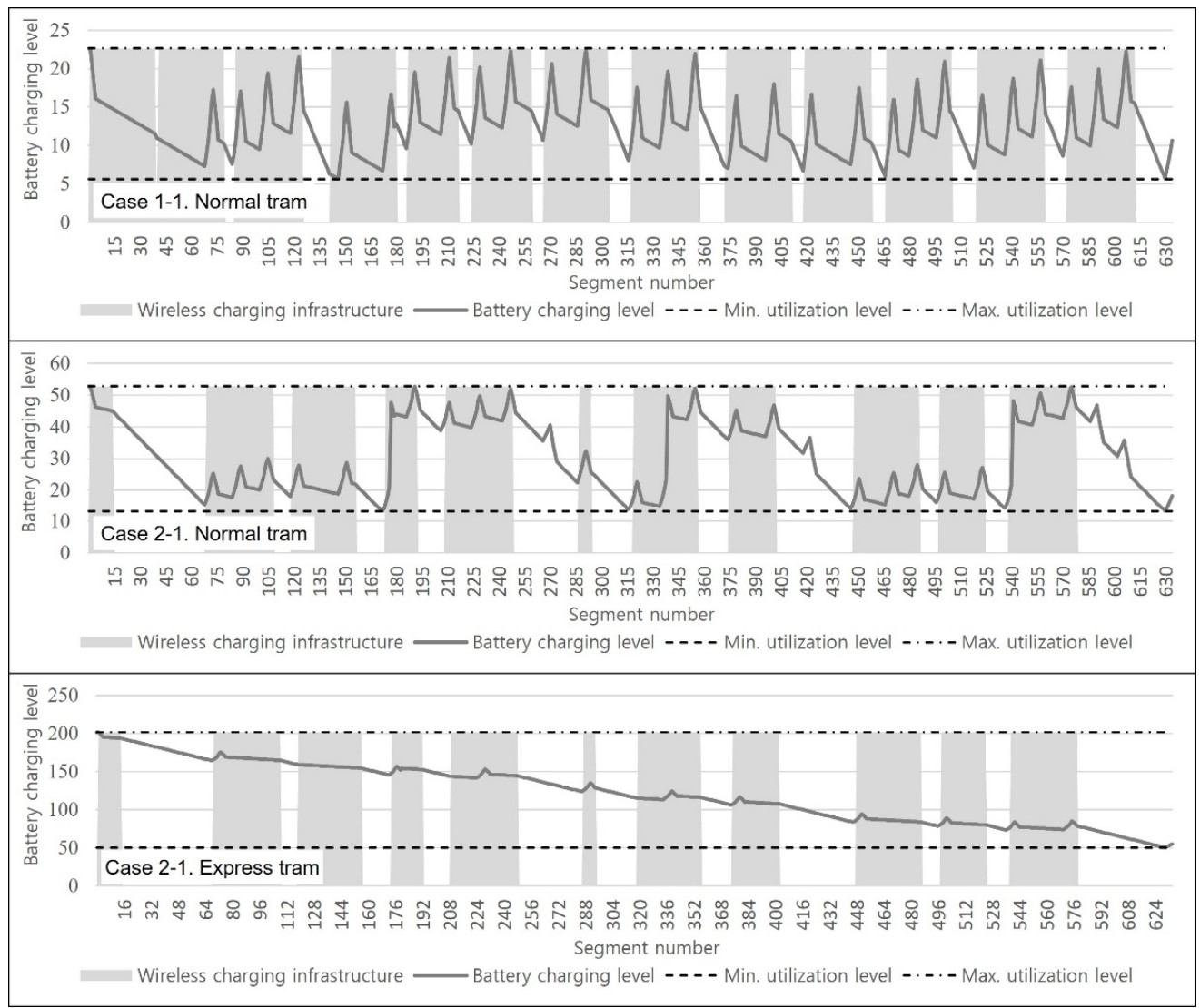

**Figure 4.** Variations of the battery charging level in Case 1-1 and Case 2-1.

The variations for the battery charging level in each case occurred within the minimum and maximum battery utilization level. The trends for the battery charging level variations for the normal trams for both Case 1-1 and Case 2-1 are similar, because there is only difference between them in the operation profile. Note that the normal tram in Case 2-1 has a stopping time in order to avoid the express tram at certain stations for three minutes. However, the express tram in Case 2-1 has a different trend, because it is stopped at several predetermined stations. In addition, the battery charging level can increase or decrease when it is operating on a railway where the wireless charging infrastructure is installed according to the summation of the battery consumption by the operation and the battery charging by both the wireless charging and the regenerative braking. However, when the wireless charging electric tram is operated on a railway without the wireless charging infrastructure, the battery charging level decreases, which is due to the battery consumption by the operation, which increases due to the battery charging by the regenerative braking.

### 4.3. Sensitivity Tests

In the previous section, computation experiments are performed with the same operation ratio between the express tram and the normal tram. To confirm the variation of the numerical results, the different operation ratios of both the express tram and the normal tram are examined. For the total of 20 operations, the variation of decision variables—the

required battery capacity and the allocation of the wireless charging infrastructure—are observed when the express operation is performed from 18 times to 2 times with decrementing by two, while the normal operation is executed from 2 times to 18 times with incrementing by two as shown in Table 6. Note that the battery capacity in Table 6 is for unit tram.

**Table 6.** Variation of the decision variables about the different operation ratios for the dual policy.

| Operation Number | | Battery Capacity | | Length of Inductive Cable | Number of Inverters |
|---|---|---|---|---|---|
| Express | Normal | Express | Normal | | |
| 18 | 2 | 65.19 kWh | 25.50 kWh | 30,000 m | 15 |
| 16 | 4 | 92.97 kWh | 25.50 kWh | 28,000 m | 14 |
| 14 | 6 | 236.95 kWh | 65.72 kWh | 17,650 m | 10 |
| 12 | 8 | 237.70 kWh | 65.74 kWh | 17,600 m | 10 |
| 10 | 10 | 252.38 kWh | 65.97 kWh | 16,550 m | 11 |
| 8 | 12 | 253.41 kWh | 66.02 kWh | 16,500 m | 11 |
| 6 | 14 | 253.41 kWh | 66.02 kWh | 16,500 m | 11 |
| 4 | 16 | 276.85 kWh | 63.89 kWh | 15,000 m | 15 |
| 2 | 18 | 285.42 kWh | 63.89 kWh | 14,400 m | 17 |

When the ratio of the express operation is high, the wireless charging electric tram system then tends to allocate the wireless charging infrastructure much more for the installation of overall system than to determine an adoption of relatively higher battery capacity in order to provide the electricity. As the number of express trams increases, there is a tendency to adopt a lower battery capacity and install longer wireless charging infrastructure to save on increasing battery costs. At this time, the number of inverters may be reduced in some sections as long split inductive cables are connected. This means that the wireless charging infrastructure tended to deploy for the express tram rather than for the normal tram, which is due to the express tram's low chance of wireless charging. Therefore, it is observed that the battery capacity for the express tram is continuously reduced. However, even if the ratio of normal operation changes, the capacity of the battery mounted on the normal tram does not change significantly and remains between 60 kWh and 70 kWh. This is because the normal trams that stop at each station have a possibility to receive an electrical energy sufficiently from the wireless charging infrastructure, since there is no significant change in the installation length of the wireless charging infrastructure. Moreover, when the operating ratio of the express tram exceeds 80%, the total investment cost increases significantly as the battery cost of the express tram increases. Therefore, it can be seen that the battery capacities of both the express tram and the normal tram are greatly reduced while the wireless charging infrastructure is greatly expanded.

In addition, with the decision variables of the wireless charging electric tram system, the required battery capacity and the allocation of the wireless charging infrastructure can be observed differently according to the number of operations for both the express tram and the normal tram. It is tested from 36 operation times to 6 operation times, which decreased by five and is illustrated in Table 7. Note that the operation ratio between the express tram and the normal tram is the same.

**Table 7.** Variation of the decision variables about the number of operations.

| Operation Number | Battery Capacity | | Length of Inductive Cable | Number of Inverters |
|---|---|---|---|---|
| | Express | Normal | | |
| 36 | 65.19 kWh | 25.50 kWh | 30,000 m | 15 |
| 30 | 92.97 kWh | 25.50 kWh | 28,000 m | 14 |
| 24 | 163.20 kWh | 30.32 kWh | 22,950 m | 13 |
| 18 | 253.73 kWh | 66.67 kWh | 16,450 m | 11 |
| 12 | 275.97 kWh | 81.43 kWh | 14,900 m | 12 |
| 6 | 485.26 kWh | 451.54 kWh | 1150 m | 10 |

In general, the wireless charging electric tram system tries to apply a relatively large number of the wireless charging infrastructures when the number of operations increases. This means that there are more batteries in the overall system, so it can reduce the total investment cost. However, the results of this sensitivity test are derived more clearly than expected. When the number of operations is between 12 and 24, the application of the wireless charging electric tram system is sufficiently efficient. The wireless charging electric tram system tries to deploy the wireless charging infrastructure at an overall railway between 47.0% and 72.4% while reducing the required battery capacity at a reasonable level. However, when the number of operations is equal to or less than 6, the wireless charging electric tram system operates similar to a pure battery-type electric tram without the deployment of the wireless charging infrastructure. In addition, when the number of operations is equal to or greater than 30, the wireless charging electric tram system operates such as a conventional electric tram with very small capacity battery and a fully deployed power supply line at all railways. From this sensitivity test, the insight about the situation to deploy the wireless charging electric tram system with efficient ways can be derived, and more intensive studies will be conducted as the further research.

## 5. Concluding Remarks

The purpose of this study is to derive the required minimum battery capacity and the allocation of the wireless charging infrastructure for the wireless charging electric tram system. The dual operation policy, which includes express and normal operations, is considered while minimizing the total investment cost. A general introduction regarding the wireless charging electric tram system as well as a literature review that is related to both the wireless charging electric tram system and the skip-stop operation are provided for the potential readers. In order to solve the presented problem, a mathematical model is developed and solved using CPLEX, which is the optimal solution generation software.

In order to check if the devised model can derive a proper result, computational experiments were conducted. However, the transportation circumstance can be different, so deriving an exactly quantitative result might not be as valuable as expected. Therefore, sensitivity tests are presented in order to derive several insights about the efficient way to deploy the wireless charging electric tram system that considers the dual operation policy. The pros and cons of adopting a battery with a larger capacity or a longer infrastructure were determined by comparing the result of the sensitivity analysis.

When the express operation and the normal operation are performed simultaneously, the wireless charging electric tram system then tends to apply greater battery capacity as opposed to only performing the normal operation. This is because the wireless charging infrastructure cannot be deployed adequately for both the express and the normal operations. Therefore, a higher total investment cost is also required under the dual operation policy, but the exact quantity can be different according to the system's circumstances. In addition, the battery charging level is maintained between the minimum utilization level and the maximum utilization level of the battery. It was observed from the sensitivity tests that the battery capacity tended to decrease, whereas the allocation of the wireless charging infrastructure is extended for all railways when the ratio of the operation of the express tram increases to save on increasing battery costs. In addition, according to

the operation number of both the express and the normal trams, it is confirmed that an adequate form of the electric tram can be different. When the number of operations is small, the pure battery-type electric tram is then a more efficient form, whereas when the number of operations is large, the conventional electric tram with fully deployed power supply line at all railways is then a more adequate form.

More various circumstances are required for the future studies in order to find the efficient conditions in regard to applying the wireless charging electric tram system. It will be possible to diversify the types of inverters and set different lengths of inductive cables that can be covered accordingly. In addition, it will be necessary to include the electrical energy required to operate the wireless charging electric tram system as a cost and to consider energy efficiency which has been overlooked in this study. Moreover, the optimal operation profile and operation schedule determination of electric trams considering the installed wireless charging infrastructure and battery capacity will need to be studied.

**Author Contributions:** Conceptualization, Y.D.K.; methodology, Y.K.K.; software, Y.K.K.; validation, Y.O. and D.Y.R.; formal analysis, Y.K.K.; investigation, Y.K.K. and D.Y.R.; resources, Y.D.K.; data curation, Y.D.K.; writing—original draft preparation, Y.K.K., Y.O. and D.Y.R.; writing—review and editing, Y.D.K.; visualization, Y.O. and D.Y.R.; supervision, Y.K.K.; project administration, Y.K.K.; funding acquisition, Y.K.K. All authors have read and agreed to the published version of the manuscript.

**Funding:** This research received no external funding.

**Institutional Review Board Statement:** Not applicable.

**Informed Consent Statement:** Not applicable.

**Conflicts of Interest:** The authors declare no conflict of interest.

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
