# Peer review of "Optimal Deployment of Wireless Charging Infrastructure for Electric Tram with Dual Operation Policy"

_vehicles, doi:10.3390/vehicles4030039_

Round 1

Reviewer 1 Report

The article addresses the relationship between allocation decisions for inductive cable allocation as well as inverter allocation for wirelessly charged electric trams and the size of the required batteries for the case of a tram line on which some trams operate in so-called normal mode, serving each possible stop, while some other trams operate in express mode, serving selected stops only. The paper addresses a relevant topic and the research methodology is adequate. The academic contribution of an improved version of the paper would be modest, but it would still justify a publication once several major problems had been addressed successfully, which I would like to encourage the authors to do. In the current state, my assessment of the paper is somewhere between a rejection and a major revision.

Major problems:

  1. The English needs substantial improvement. The paper definitely needs to be polished by a professional editing service. This does not only apply to grammar errors, but also to style and the often times bumpy train of thought. To be crystal-clear, this is a real problem which, in my eyes, can only be solved by a professional editing service.

  2. Citations should be either author-year (which I prefer), or numeric, but not both as in the current version of the paper.

  3. The structure of the paper has a major, critical flaw: In Section 3, a model is developed before the underlying problem has been clearly explained, using a minimal example which exhibits all the relevant aspects of the problem. I suggest to introduce a further section between current sections 2 and 3, so that the correctness and completeness of the model can be assessed.

  4. The model notation is poorly organized. First of all, please do not operate with indices in parentheses and as subscripts. Just use subscripts, separated by commas, if there is any ambiguity.

  5. In your notation, battery levels are denoted as parameters. However, they are decision variables.

  6. In Table 6, neither the meaning nor the dimensions of the parameters are given.

  7. The results in Table 4 contradict those in Figure 4 as the battery capacities in Case 2 in the table seem to messed up.

  8. I do not really understand the result in Table 5. You seem to suggest that if a single inverter is connected to an inductive cable of a length of 30,900 meters, almost the entire length of the tram line, a giant battery capacity for the normal tram is required. I would expect that more inductive cable leads to smaller battery sizes, as it does for the express tram.

  9. Is is really realistic to have a single cable of 30,900 meters connected to a single inverter, which then provides the power for all the trams at the same time? Is this technically feasible?

  10. Your notation of the “number of operations” in Table 6 is unclear. Are you really considering to have 50 trams simultaneously operating on those 31 km of line length? Is this a realistic or interesting case?

  11. Please report the CPLEX solution times and MIP gaps for the model instances.

Minor problems:

  • Lines 38-39: Sentence is unclear.

  • Line 72: … decision about the location of the express stations …

  • Line 96: The battery is expensive, not the battery price.

  • Line 204+: Indices and index sets!

  • N_k is dimensionless!

  • Line 206: No, the purpose of the model is not to generate anything. The purpose of the model is to clearly define the problem.

  • P. 6: Do not use braces in the objective function.

  • Line 244: What is a reasonable time?

  • Table 3: How is it that the tram vehicle has zero cost? If it has, why include it in the model?

  • Table 4: Are the numbers about the battery cost for all trams together?

  • Line 304: Shouldn’t it be 0.9 or 90%, but not 0.9%?

  • Table 5: Are the battery capacities those for a single tram or all trams of a given type together?

Author Response

Please find an attached file.

Reviewer 2 Report

In this paper, the authors propose an optimal deployment of the wireless charging electric tram system considering both express and normal operation policies based on the Station information of Metro 9 in Seoul Metropolitan City. The objective is to minimize the total investment cost. 

However, in my opinion, the paper has some shortcomings regarding data analyses and text, and I feel this work has not been utilized to its full extent. Moreover, the paper is not satisfying, well-written, and organized; in the current form, it needs a deeper discussion on the contribution of the results. The English writing and structure need improvement; besides, better and more related citations are needed to help the authors and provide better and more accurate information. A major revision is suggested to be possibly acceptable in this journal publication. 

The following points could be used to improve this paper further:

1- The introduction is not comprehensive to give recent development in the considered field. Besides, it is not written with an organized trend. 

2- In the introduction, the contribution is not well satisfying. What achievements of this paper make it valuable to be published compared to other similar and related review papers? 

3- In the introduction, the authors stated that: the battery is environmentally harmful material". It must be clearly described how and to what extent this charging pattern addresses this problem?

4- In defining the problem of minimization of the total investment cost, has the factor of energy loss been considered?

5- What factors influence the maximum and minimum battery utilization ratios in equations 2 and 3?

6- There must be definitions for the parameters in Table 2.

7- In Figure 4, there is no symbol for the measuring unit on the y-axis.

8- The conclusion part is too long. It should provide the main points and results.

Author Response

Please find an attached file.

Reviewer 3 Report

The writer should consider the following publication.

Nahum, O. E., & Hadas, Y. (2020). Multi-Objective Optimal Allocation of Wireless Bus Charging Stations Considering Costs and the Environmental Impact. Sustainability, 12(6), 2318.

I have some remarks that the author should address.

1. Batteries usually come in pre-defined sizes, therefore solutions such as 29.42kWh or 78.52kWh, make no sense. The Author should change its mathematical formulation, such that it will only be possible to select a battery from a set of batteries.

2. The same applies to the inductive cable.

3. It is not clear to me, and perhaps the author didn't consider it, does the author consider physical limitations along the path. It is possible that the model will suggest installing an inductive cable of a certain length and a certain position, however, physically it is not possible.

Author Response

Please find an attached file.

Round 2

Reviewer 1 Report

I acknowledge that you made a serious effort to improve your paper by seriously considering my questions and suggestions from the first review. I have two final comments:

1. On page 6, you make a distinction between "Variables" and "Decision variables". From a mathematical point of view, this is not convincing, as both are unknowns to be determined in the solution process. So "Imax,k" and "Imin,k" are actually decision variables, because they are linearly tied to the decision variable q_battery,k. Furthermore, the battery charging levels I_i,k are also decision variables. I suggest to correct the presentation of the notation accordingly.

2. If you decide to operate with the numerical system for the references, then the way how you construct your sentences should be modified. So in line 124 I would not write "[5] recently derived ...." but rather "Hwang et al. [5] recently derived ... " It is just more natural to read this way. I suggest to use author names in the sentences instead of numbers when they serve as the subject of a phrase.

Author Response

Reviewer Comments and Author Responses

2nd Round Revision, vehicles-1770599

I appreciate all the efforts and supports of referees on the evaluation and important comments for my manuscript. In the revised version, I tried to sincerely handle every concern and suggestion raised by referees. Please refer following responses.

Referee #1

Q1. I acknowledge that you made a serious effort to improve your paper by seriously considering my questions and suggestions from the first review. I have two final comments:

  1. On page 6, you make a distinction between "Variables" and "Decision variables". From a mathematical point of view, this is not convincing, as both are unknowns to be determined in the solution process. So "Imax,k" and "Imin,k" are actually decision variables, because they are linearly tied to the decision variable q_battery,k. Furthermore, the battery charging levels I_i,k are also decision variables. I suggest to correct the presentation of the notation accordingly.

Responses: Thank you for your valuable comments. As you mentioned, I redesigned of notation table as follow.

Decision variables

xi

:

Binary decision variable, which has a value of 1 when the inductive cable is allocated at the ith segment, otherwise, it has value of 0

yi

:

Binary decision variable, which has value of 1 when the inverter is allocated at the ith segment, otherwise, it has value of 0

qbattery,k

:

Minimum required battery capacity for the wireless charging electric tram, which is operated by type k [kWh]

Imax,k

:

Maximum utilization level of a battery for the type k electric tram [kWh]

Imin,k

:

Minimum utilization level of a battery for the type k electric tram [kWh]

Ii,k

:

Battery charging level after passing the ith segment for the type k electric tram [kWh]

Q2. If you decide to operate with the numerical system for the references, then the way how you construct your sentences should be modified. So in line 124 I would not write "[5] recently derived ...." but rather "Hwang et al. [5] recently derived ... " It is just more natural to read this way. I suggest to use author names in the sentences instead of numbers when they serve as the subject of a phrase.

Responses: Thank you for your adequate comments. All literature citations in my manuscript have been changed according to your suggestion. Please check the revised manuscript.

This manuscript is improved to have more contributions because of the reviewer's insightful advice. I really appreciate your advice.

Reviewer 2 Report

The authors addressed the concerns well. The paper can now be accepted for publication.

Author Response

Reviewer Comments and Author Responses

2nd Round Revision, vehicles-1770599

I appreciate all the efforts and supports of referees on the evaluation and important comments for my manuscript. In the revised version, I tried to sincerely handle every concern and suggestion raised by referees. Please refer following responses.

Referee #2

Q1. The authors addressed the concerns well. The paper can now be accepted for publication.

Responses: Thank you for your valuable comments. This manuscript is improved to have more contributions because of the reviewer's insightful advice. I really appreciate your advice.
